# A Physiological and Molecular Focus on the Resistance of “Filippo Ceo” Almond Tree to *Xylella fastidiosa*

**DOI:** 10.3390/plants13050576

**Published:** 2024-02-20

**Authors:** Mariarosaria De Pascali, Davide Greco, Marzia Vergine, Giambattista Carluccio, Luigi De Bellis, Andrea Luvisi

**Affiliations:** 1Department of Biological and Environmental Sciences and Technologies, University of Salento, 73100 Lecce, Italy; mariarosaria.depascali@unisalento.it (M.D.P.); davide.greco@unisalento.it (D.G.); giambattista.carluccio@unisalento.it (G.C.); luigi.debellis@unisalento.it (L.D.B.); andrea.luvisi@unisalento.it (A.L.); 2National Biodiversity Future Center, 90133 Palermo, Italy

**Keywords:** climate change, combined stresses, drought, plant disease, transcription factor

## Abstract

The impact of *Xylella fastidiosa* (*Xf*) subsp. *pauca* on the environment and economy of Southern Italy has been devastating. To restore the landscape and support the local economy, introducing new crops is crucial for restoring destroyed olive groves, and the almond tree (*Prunus dulcis* Mill. D. A. Webb) could be a promising candidate. This work focused on the resistance of the cultivar “Filippo Ceo” to *Xf* and evaluated its physiological and molecular responses to individual stresses (drought or pathogen stress) and combined stress factors under field conditions over three seasons. Filippo Ceo showed a low pathogen concentration (≈10^3^ CFU mL^−1^) and a lack of almond leaf scorch symptoms. Physiologically, an excellent plant water status was observed (RWC 82–89%) regardless of the stress conditions, which was associated with an increased proline content compared to that of the control plants, particularly in response to *Xf* stress (≈8-fold). The plant’s response did not lead to a gene modulation that was specific to different stress factors but seemed more indistinct: upregulation of the *LEA* and *DHN* gene transcripts by *Xf* was observed, while the *PR* transcript was upregulated by drought stress. In addition, the genes encoding the transcription factors (TFs) were differentially induced by stress conditions. Filippo Ceo could be an excellent cultivar for coexistence with *Xf* subps. *pauca,* confirming its resistance to both water stress and the pathogen, although this similar health status was achieved differently due to transcriptional reprogramming that results in the modulation of genes directly or indirectly involved in defence strategies.

## 1. Introduction

In nature, plants face numerous abiotic and biotic stress factors simultaneously, which increase due to climate change, thus affecting their growth, yield and survival. Among the possible stress combinations, the combination of drought and pathogens is one of the most devastating [1,2]. Drought is a frequent environmental stress that exacerbates the damage caused by bacterial pathogens [3,4]. For example, drought worsens Pierce’s disease caused by *Xylella fastidiosa* (*Xf*) in *Vitis vinifera* [5], the bacterial blight caused by *Xanthomonas oryzae* pv. *oryzae* in rice [6], and the common scab disease caused by *Streptomyces* spp. in potato [7]. However, other works have reported that drought acts positively, improving the plant defence response against pathogens, as reported in tomato and grapevine against the necrotrophic fungus *Botrytis cinerea* [8,9]. Several studies [10,11,12,13] have also shown that Leccino’s resistance to *Xf* may be attributed to its vulnerability to water deficit. This susceptibility could activate alternative defence mechanisms that assist the plant in response to pathogens.

To date, the relationship between the mechanisms of host resistance and drought tolerance has not been determined. At the molecular level, the only certainty is that both are controlled by a complex network of events involving the activation/inactivation of the expression of a large number of genes [14]. In particular, drought stress induces changes in the activity of genes encoding stress response proteins, including dehydrins (DHNs) and late embryogenesis-abundant (LEA) proteins, which enable plants to resist drought [15]. On the other hand, phytopathogens primarily induce the expression of genes encoding pathogenesis-related (PR) proteins [16], which are the molecules recruited to defend plants against pathogen attack and are also involved in the crosstalk of abiotic and biotic stress signalling.

In recent years, transcription factors (TFs) have been established as the main regulators of changes in gene expression and, thus, the major factors that facilitate stress responses in plants. In fact, TFs control the transcription rate by binding to cis-regulatory promoter elements and play a significant role in signal transduction networks. This thus leads to an improvement in plant tolerance.

In plants, more than 80 TF families have been identified. Some of these proteins, such as basic region-leucine zipper (bZIP), myeloblastosis-related proteins (MYB), NAC, WRKY and Zn finger proteins, are directly implicated in stress responses and are associated with enhanced resistance. In addition, they have been reported to impart plant cross-tolerance to abiotic and biotic stresses [17]. For instance, *Os*bZIP23 overexpression was shown to confer abscisic acid (ABA) hypersensitivity and increased salinity and drought tolerance in rice [18], and NAC TFs were shown to directly induce the pathogenesis-related genes *PR1*, *PR2*, and *PR5* [19] and enhance drought resistance [20]. WRKY proteins are particularly associated with the regulation of plant pathogen responses. However, recent functional analyses have also implicated WRKY TFs in abiotic stress responses [21]. Several studies have focused on the role of MYB TFs as key factors in regulating abiotic and biotic stress responses. As reviewed by Fang et al. [22] MYB TFs are active in stress signalling because they regulate downstream genes in response to stresses. Finally, Zn Finger enhances plant drought resistance by increasing the levels of osmotic adjustment substances. In fact, overexpressing *OsMSR15* [23] and *ZFP3* in transgenic *Arabidopsis* [24] results in an increase in drought tolerance by maintaining a higher proline content, reducing electrolyte leakage, and increasing stress-responsive gene expression. Generally, these TFs play a crucial role in abiotic and biotic stress responses [25]. Their ability to control a set of genes to modulate their expression via different pathways in response to various stimuli empowers plant defence. TFs are considered excellent targets for increasing plant adaptation to stress.

*Xylella fastidiosa* (*Xf*) is one of the most devastating pathogens and is able to infect a wide range of host plants, causing diseases that can cause severe yield losses in highly economically important crops, such as Pierce’s disease in grapevine, citrus variegated chlorosis, olive quick decline syndrome (OQDS) and almond leaf scorch disease (ALSD) [26]. The latter is a severe disease that threatens almond (*Prunus dulcis* Mill. D. A. Webb) in several areas worldwide [27,28,29]. Recently, the pathogen caused severe yield losses in almond crops and eradicated 1000 trees in Spain. ALSD has also affected more than 79% of almond trees in Majorca [30]. Since 2017, symptoms of ALSD have also been observed on 30-year-old almond trees in mainland Spain [31]. According to Amanifar et al., 2022 [32], the severity of the disease is also related to the sensitivity of the cultivars.

Since 2013, *Xf* subsp. *pauca* has caused OQDS in Apulia and destroyed millions of olive trees, leading not only to massive damage to the local economy but also to complete changes in the landscape. In addition to olive trees, which exhibit lower infection levels than infected olive trees, almond trees were also found to be infected and symptomatic [33]. Although these features have not been fully investigated, they make almond trees effective for carrying out crop renewal to create new production chains and increase the biodiversity in an area entirely ravaged by the bacterium [34,35].

The almond tree has adapted to temperate and Mediterranean regions, as it can grow in conditions of water shortage without requiring irrigation, owing to adaptive mechanisms such as osmotic adjustment, stomatal conductance decreases, and the water loss rate increases [36]. In addition, the almond tree is an icon for the agricultural landscape of Apulia and is of significant economic importance. “Filippo Ceo” is the most appreciated cultivar in Apulia due to its high productivity and fruit yield. It has also been shown to have resistance traits that are similar to those of resistant olive tree varieties [35]. Considering the projections of the increasing impact of climate change, this work focuses on the resistance of “Filippo Ceo” to *Xf*. The study also evaluates the individual and combined effects of pathogens and water deficit to contribute to plant management and protection in areas threatened by *Xf.*

We thus evaluated the response of the “Filippo Ceo” cultivar to *Xf* infection and drought under individual stress (drought or pathogen) and combined stress in field conditions in a two-year trial. We assessed the changes in physiological parameters and studied the expression profiles of genes encoding proteins (DHN, LEA, and PR) and TFs (bZIP, MYB, NAC, WRKY, and Zn finger) involved in the response to these stresses. The overall aim of the study is to provide information on the resistance mechanisms of the almond tree Filippo Ceo for use in the recovery of an area compromised by the devastating action of the pathogen.

## 2. Results

### 2.1. Plant Health and Physiological Characterisation

No trees in the orchard under investigation experienced symptoms associated with *X. fastidiosa* infection throughout the whole trial period (2021–2022). Regarding the *Xf*-positive plants, the PCR-based analyses showed values not exceeding 10^3^ CFU mL^−1^ (Figure 1). Thus, the bacterial concentration detected was rather low, which probably explains the absence of symptoms.

The physiological characterisation of the cv. “Filippo Ceo” was performed by evaluating the relative water content (Figure 2). During the two seasons of the study, the control plants presented constant RWC values (RWC 95%), indicating an optimal and stable water status regardless of the sampling period. Under individual stress and in combination with drought/*Xf*, the RWC values slightly decreased compared to those of the control plants. This reduction did not exceed 10%, thus maintaining good water status, as shown by the high RWC values (82–89%). Notably, within each sampling period, the different stresses led to a substantially similar RWC, indicating that the water status of the leaves was essentially the same both when the plants were not irrigated and in the presence of the pathogen or when the two factors were combined.

Compared with those of the control, the stress conditions significantly increased the content of free proline (Figure 3). Water deficit led to similar proline accumulation patterns regardless of the sampling period, with an average 3.46-fold increase compared to that of the control. Most strikingly, more proline accumulated in response to *Xf*. Our data showed an average increase of 8.14-fold compared to the control conditions. This trend was confirmed in all sampling seasons. The simultaneous action of drought and pathogen led to an average 6.43-fold increase compared to the non-stressed plants, higher than the water stress but lower than the pathogen alone.

### 2.2. Gene Expression Analysis

To gain insights into the molecular mechanisms potentially involved in the resistance of “Filippo Ceo” to individual and combined stresses, we assessed the expression patterns of several marker genes encoding proteins and transcription factors known to be involved in drought and pathogen stress responses. The plant’s response to different stress factors did not lead to specific gene modulation; however, the response was less clear.

*Xf* induced the expression of genes associated with abiotic stress the most (DHN and LEA), just as water stress better stimulated the expression of genes related to biotic stress (PR) (Figure 4). In fact, *Xf* significantly increased the expression of the LEA and DHN genes in spring and summer (e.g., up to 2.48 log_2_ FC for LEA and up to 2.71 log_2_ FC for DHN). Although these genes play a crucial role in plant adaptation to drought stress, the pathogen leads to their expression, suggesting their involvement in plant defence strategies. Similarly, the PR gene, which is usually known to be involved in the pathogen defence mechanism, exhibited a greater response to drought, regardless of the season. This trend was most evident in the spring when the relative expression level was 2.16 log_2_ FC. In contrast, the combination of abiotic and biotic stress did not result in differential expression of the three genes across the seasons considered, with similar or intermediate values to the single stress factors in spring or lower values in the rest of the year.

Concerning the expression profiles of the genes encoding TFs (Figure 5), all the genes were induced under drought, although with generally lower expression than that of *Xf* or the combined stress factors. The only exceptions were observed in spring for bZIPs and, to a lesser extent, for Zn fingers because a greater response of bZIPs (4.63 log_2_ FC) was found together with a significant accumulation of transcripts for Zn fingers. The genes *MYB*, *NAC* and *WRKY* were not particularly affected by drought, and a very low expression level was observed for all the TFs analysed in the summer and autumn. The expression profile of the pathogen stress factor was completely different from that under the other stress conditions. With the exception of the aforementioned divergence for *bZIP* and *Zn fingers* in spring, *Xf* seemed to stimulate greater expression of TFs than water stress alone in all the seasons. This trend was most evident in the spring when the pathogen induced the expression of *MYB*, for which the value was 3.69 log_2_ FC, while the values of *NAC*, *WRKY*, *bZIP* and *Zn finger* were between 2.49 and 1.84 log_2_ FC. In the summer, the expression levels of all the genes considered were similar (range between 1.47 and 1.87 log_2_ FC), while in autumn, the expression level was lower than 1 log_2_ FC. *MYB* (2.26 log_2_ FC), *WRKY* (2.42 log_2_ FC), and *Zn finger* (2.22 log_2_ FC) genes were notably upregulated in response to the combined stress factors in the spring. A notable decrease in gene expression was observed in the summer and autumn, with values below 1 log_2_ FC, except for *MYB*, which showed an expression level equal to 1.60 log_2_ FC only in the summer. As with gene expression, combined stress factor-related stress sometimes led to similar or intermediate TF expression levels compared to those resulting from single stress.

## 3. Discussion

In Salento, the spread of *Xf* has significantly damaged both the environment and the economy. This has led to the development of strategies aimed at restoring not only the landscape but also bolstering the local economy that has been severely affected by OQDS. One of the most effective strategies could be the varietal renewal of destroyed olive groves, not only through the use of cultivars that are resistant to bacteria but also through the introduction of new crops. Due to some of its particular characteristics, the “Filippo Ceo” almond tree could be an excellent candidate because it does not require frequent irrigation, and no appreciable symptoms attributable to *Xf* were observed during the trial, despite the high inoculum pressure present in the area. This finding suggested that “Filippo Ceo” is a cv resistant to the *Xf* subsp. *pauca* strain “De Donno”.

We thus investigated the mechanisms underlying this ability. The RWC values measured were high both for the control plants and for the stressed plants, confirming that this almond cultivar is a hardy plant that is capable of resisting adverse factors. In fact, plants that maintain an excellent physiological balance under stress conditions have higher RWC values. In contrast, plants with lower RWC values are believed to be more sensitive to water deficit, making the RWC an excellent indicator of the plant’s water status and resistance to water stress. In this study, the analysed “Filippo Ceo” almond plants exhibited consistent relative water content (RWC) values and health statuses, despite the different biotic, abiotic, or combinations of stressors. This difference may be due to distinct resistance mechanisms, possibly implicating proline accumulation as a contributing strategy. Proline stabilises the cellular structure, proteins, and enzymes, acts as an antioxidant and provides ROS defence [37]. It thus works as an osmoprotectant, enabling plants to tolerate stress [38]. By accumulating proline, plants lower their osmotic potential and delay drought-responsive stomatal closure through turgor maintenance and sustain normal photosynthesis and assimilation, thus maintaining plant growth and development [39]. Our findings showed elevated proline levels in almond trees subjected to stress conditions. In particular, the presence of the pathogen led to a more significant increase in the production of proline than that in the control, drought, or combined stress conditions, despite the similar RWC values. This finding suggested the active role of proline in the resistance of “Filippo Ceo” to *Xf*. However, further studies are required to fully understand the relationship between solute accumulation and stress adaptation.

At the molecular level, stress triggers cascading events that culminate in gene expression changes. It is widely reported that dehydrins and LEA proteins play a primary role in the response of plants to abiotic stress. They perform specific protective functions in plant cells, such as maintaining the integrity of crucial cell structures, and alleviate oxidative damage in stressed plants. In this work, the *DHN* and *LEA* genes were induced by all the stressors considered, confirming their involvement in stress responses. However, unexpectedly, *Xf* presented the greatest accumulation of transcripts compared to those under drought and combined stress. The roles of DHNs and LEA proteins in response to abiotic stress are well established, but their involvement in biotic stress responses is relatively unknown. However, some studies have shown that *DHNs* can be induced in response to attack by filamentous pathogens, such as *Erysiphe necator* [40], in grapevine plants or in combination with drought stress, as observed in oak plants infected with *Phytophthora cinnamomi* [41]. In olive, *DHNs* have also been shown to be induced by *Xf* in combination with drought [11]. Our results for almond suggest putative roles for DHNs and LEA proteins in modulating defence responses to vascular pathogens. These findings suggest that stress-related proteins may play a fundamental role in protecting plants against biotic stress.

In contrast, the *PR* genes are commonly induced by phytopathogens as well as defence-related signalling molecules, leading to increased resistance to pathogens [11,42,43]. Recent studies have reported that *PR* genes are also significantly induced by abiotic stressors, which makes them highly promising candidates for developing crop varieties that can tolerate multiple stresses [44,45,46]. In fact, our data showed that the transcript levels of *PR-1-like* were greater in response to drought stress than in response to *Xf* infection, confirming that these genes are activated not only in response to pathogen attack. Taken together, these data suggest significant crosstalk and trade-offs between almond tree responses to water deficit and *Xf*. In fact, the plant’s response to different stress factors did not lead to specific gene modulation; however, the response appeared to be more unclear. This finding thus confirms that stressor responses share protective mechanisms (cross-tolerance) or signalling/regulatory pathways that activate independent protective mechanisms (cross-talk) [47].

TFs are being extensively studied because, similar to switches, they monitor the activity of stress responses in many genes in a coordinated manner and represent tools for enhancing abiotic or biotic stress tolerance in plants. TFs play a significant role in signal transduction networks, from the perception of a stress signal to the regulation and expression of almost any gene [48]. This work analysed the expression profiles of genes encoding transcription factors such as MYB, NAC, WRKY, Zn finger and bZIP. In particular, MYB proteins support a wide range of signalling cascades between abiotic and biotic stress signals [49,50]. The *MYB* gene also controls the production of dehydrins and LEA proteins, along with a greater accumulation of sugars and proline, and in *Vitis vinifera,* upregulation of the MYB transcription factor was observed in response to *Xf* [51]. In our work, the *MYB* gene was induced by all the stress factors considered; however, compared with drought and combined stress factors, *Xf* led to the greatest accumulation of transcripts during the period of greatest vegetative development. The same trend was observed for *DHN* and *LEA* gene expression, suggesting that MYB may be involved in their activation.

However, in the spring, the expression of the *ZIP1* gene increased in response to a water deficit compared with that in response to *Xf* and the combined stress factors. ZIP proteins control the signal transduction networks that mediate the response to drought [52].

In response to pathogen infection, numerous *NAC* genes are induced [53], and the overexpression or silencing of these genes results in enhanced or reduced resistance to pathogens [54]. In line with findings reported in the literature, the *NAC* gene is mainly induced by the pathogen, confirming that NAC TFs link signalling pathways to regulate resistance against pathogens. However, upregulation of the transcription factor NAC was recently not detected in the *Xf*-infected almond cv. Avijor [55], suggesting a different response of the cultivars to ALSD symptoms [26].

In the spring, the *WRKY* transcript level increased significantly in response to the combination of water deficit and pathogen stresses. Several WRKYs are active at the crossroads of plant responses to biotic and abiotic stresses [56]. In particular, a study conducted by Lee et al. in 2018 [57] demonstrated that OsWRKY11 serves as a positive regulator of plant defence responses against pathogen infection and drought stress.

Zn finger motifs (ZFPs) are crucial for plant growth and development, stress tolerance, transcriptional regulation, RNA binding and protein–protein interactions [58]. Several studies have reported that ZFPs play a significant role in the abiotic stress response in plants [59]. In particular, several ZFPs have been shown to play significant roles in enhancing drought [60]. In our work, however, drought stress led to a significant increase in the transcription of genes only in the spring, probably due to concomitant vegetative development and high water requirements. According to the gene expression profile results, the ZFP TF seems to be more involved in the pathogen response. Several studies have shown the involvement of ZFPs in plant–pathogen interactions. For example, the overexpression of these genes in transgenic tobacco plants has been found to enhance immunity against pathogens and induce the expression of defence-related genes [61]. ZFPs may thus also be involved in resistance to *Xf*.

Generally, the transcription factors analysed presented different expression profiles and were induced by all the stress factors in the spring. However, in the summer and autumn, the presence of the pathogen alone led to a significant accumulation of transcripts. These findings suggest the putative importance of these genes in the resistance of almond trees to *Xf*.

## 4. Materials and Methods

### 4.1. Plant Materials

The study was carried out in an 18-year-old commercial almond orchard located in Veglie (Lecce, Italy) during the 2021 and 2022 seasons. The experiments were conducted on the Apulian variety “Filippo Ceo” grafted onto GF-677 rootstock.

The planting layout consisted of trees distributed in 16 horizontal rows and a planting distance of 6 m × 4 m on sandy soil (average soil texture parameters: 78% sand, 15.4% silt, 5.1% clay, and 1.5% organic matter).

The orchard is located in an area where *Xf* has been present since 2015 [61]. One municipality within the area was declared to be infected in 2015 and overwhelmed by the pathogen. The orchard is thus subjected to the continuous pressure of the natural inoculum of *Xf,* as it is surrounded by olive groves that have been seriously affected by OQDS.

The experimental design followed a randomised block plan, and each experimental set consisted of three trees (in total, n = 12 trees per treatment). Sampling was carried out at three different climatic stages (spring, summer and autumn). The experimental design included four plant conditions: *Xf*-positive trees naturally infected and irrigated (“*X. fastidiosa*”, three plants/cultivar); *Xf*-negative trees subjected to water deficit (“drought”, three plants/cultivar); *Xf*-positive trees subjected to water deficit (“combined”, three plants/cultivar); and *Xf*-negative trees and irrigated (“control”, three plants/cultivar).

The selected almond trees had previously received the same agronomic treatments. The insect control and phytosanitary treatments, according to EU Decision 2015/789 [62], were carried out by the farmers. In addition, the trees were monitored for symptoms caused by natural infection with *Pseudococcus viburni*, *Pseudomonas siringae*, *Xanthomonas arboricola* pv. *pruni*, and *Candidatus Phytoplasma phoenicium* during sampling. Plants showing symptoms related to potential co-infections were excluded from the trial. Diagnostic tests (real-time PCR) for detecting Plum pox virus, according to Olmos et al. [63], were also carried out.

The *Xf*-positive or *Xf*-negative plants were assessed for the presence of symptoms using the severity scale of 1 to 3, proposed by Luvisi et al. [64] and the qPCR assay according to Harper et al. [65]; plants were tested in 2021 and in 2022 during the three sampling periods (spring, summer, autumn). The plants were considered *Xf*-negative when the twig samples were negative according to the *Xf* assay in each sampling period and positive when the twig samples of the Filippo Ceo trees were positive according to the *Xf* assay, with Cq values ≤ 32 in each sampling period. The *Xf* concentration, expressed as bacterial CFU mL^−1^, was inferred from Cq values using a standard curve with dilutions ranging from 10^2^ to 10^7^ CFU mL^−1^, as described by D’Attoma et al. [66].

The trees were subjected to two irrigation regimens. The “*X. fastidiosa*” and “Control” plants were watered weekly following local practices, maintaining at least 90% of the soil water capacity (SWC). For the “Drought” and “Combined” plants, the regime soil moisture was maintained at approximately 40% of the SWC. The distance between blocks of irrigated and non-irrigated plants was at least 10 m.

### 4.2. Relative Water Content (RWC)

To determine the leaf water status, ten leaves per tree were placed in tubes, and the tubes were closed on site. In the laboratory, several parameters were analysed: fresh weight (FW), turgid weight at full turgor (TW) (measured after the leaf petioles were immersed for 24 h in deionised water at 4 °C), and dry weight (DW) (measured after drying at 80 °C). The RWC was calculated as follows: RWC (%) = (FW − DW)/(TW − DW) × 100. The RWC measurements were performed in spring (May), summer (July), and autumn (September) in 2021 and 2022.

### 4.3. Free Proline Determination

A total of 0.5 g of almond leaves from control and stressed plants was homogenised in 10 mL of 3% aqueous sulfosalicylic acid to determine the free proline content. The proline concentration was calculated according to Bates et al. [67].

### 4.4. Total RNA Isolation, cDNA Synthesis, and Real-Time PCR Analysis

Total RNA was extracted from leaves via the CTAB-based procedure according to the methods of Gambino et al. [68]. RNA samples were treated with DNase I (Promega, Madison, WI, USA), after which the absorbance was read at 260 and 280 nm to determine the RNA concentration and purity. According to the manufacturer’s instructions, cDNA synthesis was performed using TaqMan^®^ Reverse Transcription Reagents (Applied Biosystems, Waltham, MA, USA) with oligo (dT)18 as a primer. RT–qPCR was carried out using SYBR Green fluorescent detection in a real-time PCR thermal cycler (QuantStudio™ 3 Real-Time PCR System, Applied Biosystems, Waltham, MA, USA). The PCR program was as follows: 2 min at 50 °C and 10 min at 95 °C, followed by 45 cycles of 95 °C for 15 s and 60 °C for 1 min [12]. The primers used (Table 1) were retrieved from the literature. The primers used were designed for genes related to drought responses, such as DHN and LEA; for genes involved in the pathogen stress response, namely, pathogenesis-related protein 1-like (PR); and finally, for genes encoding TFs such as bZIP, NAC, MYB, WRKY, and Zn Finger.

To standardise the results, the relative abundance of the actin gene (Actin) was used as the internal standard (Table 1). Relative gene expression levels were calculated with the log_2_ 2^−ΔΔCt^ method [74,75]. The efficiency of the target amplification was evaluated for each primer pair, and the corresponding values were used to calculate the fold changes (FCs) with the following formula: FC = (1 + E) − ΔΔCt, where ΔΔCt = (Ct_target_ − Ct_UBQ_)_Treatment_ − (Ct_target_ − Ct_UBQ_)_Control_.

### 4.5. Statistical Analysis

The means of the quantitative data related to the RWC, proline content and gene expression levels were determined for each season (spring, summer and autumn) and subjected to one-way ANOVA, followed by the Tukey HSD (honestly significant difference) post hoc test (*p* < 0.05). Analyses were carried out using GraphPad software, version 8.02.

## 5. Conclusions

*Xf* induces physiological conditions similar to those caused by a water deficit in plants. In this work, the “Filippo Ceo” almond plants maintained good water and health status regardless of abiotic, biotic or combined stress conditions, thus confirming their resistance to both water stress and the pathogen and the combination of both. However, these similar health statuses were achieved differently because of differences in proline accumulation and differential gene expression profiles during the pathogen response. These data suggest that almond tree resistance to *Xf* could be due to transcriptional reprogramming that results in the modulation of genes directly or indirectly involved in defence strategies.

We believe that this is the first study to explore the mechanisms underlying resistance to *Xfp* in the “Filippo Ceo” almond cultivar. This study provides a foundation for future research on identifying valuable traits to combat this pathogen in affected areas. Additionally, our work sheds light on the complex interactions among plant responses to multiple stress conditions, which is especially important for future climate change scenarios.

## Figures and Tables

**Figure 1 plants-13-00576-f001:**
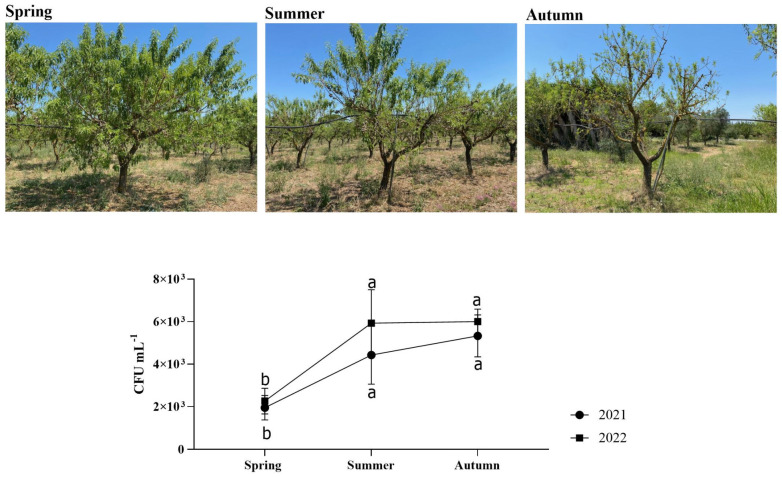
CFU mL^−1^ of *Xf*-positive almond plants sampled in this study. Statistical analysis was carried out by ANOVA followed by the Tukey HSD post hoc test. Different letters correspond to statistically different means.

**Figure 2 plants-13-00576-f002:**
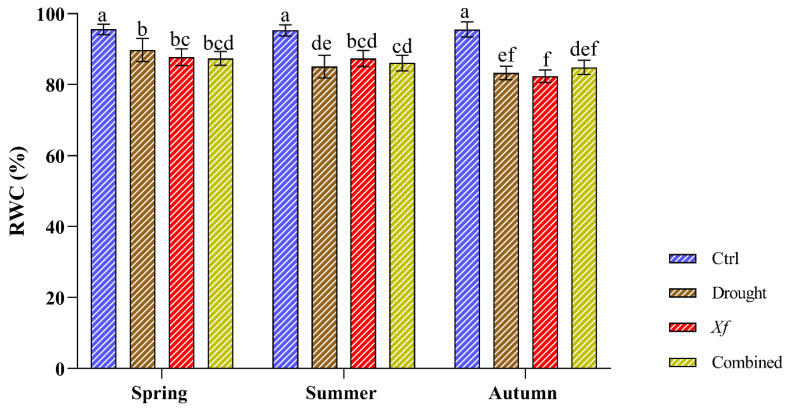
Relative water content (RWC) determined on cv. “Filippo Ceo” leaves subjected to drought, *Xylella fastidiosa* and combined over a two-year period of observation. Statistical analysis was carried out by ANOVA followed by the Tukey HSD post hoc test. Different letters correspond to statistically different means.

**Figure 3 plants-13-00576-f003:**
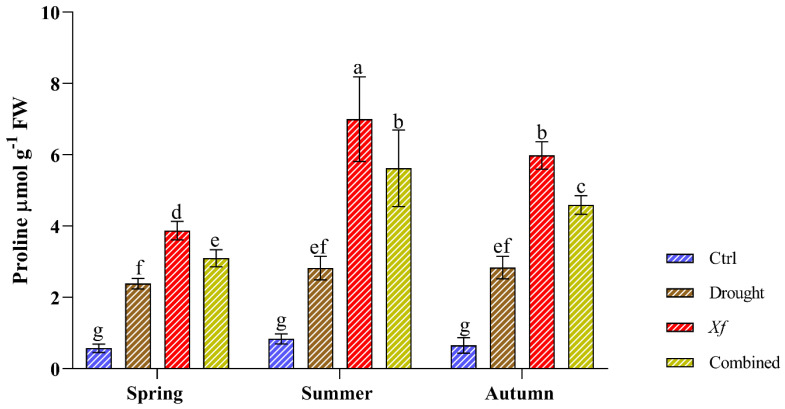
Proline content (µmol g^−1^ FW) determined in leaves of the “Filippo Ceo” cultivar subjected to drought, *Xylella fastidiosa* and combined stresses via two-year observation. Statistical analysis was carried out by ANOVA followed by the Tukey HSD post hoc test. Different letters correspond to significantly different means.

**Figure 4 plants-13-00576-f004:**
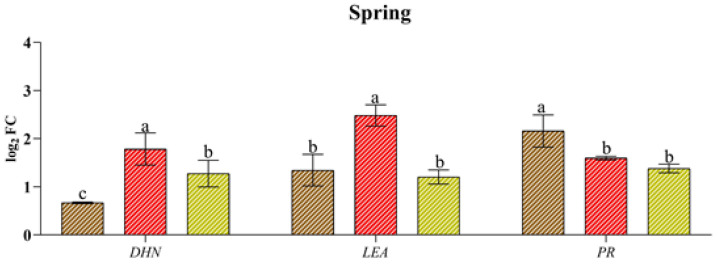
Expression analysis of dehydrin (DHN), the late embryogenesis-abundant protein (LEA) pathogenesis-related protein (PR) in leaves of cv “Filippo Ceo” subjected to different stress factors: drought, pathogen *Xylella fastidiosa* and a combination of both (spring, summer and autumn) in two-year observations, expressed as log_2_ fold change (log_2_ FC). One-way ANOVA with Tukey’s HSD post hoc test was used for the statistical analysis. Different letters correspond to significantly different means.

**Figure 5 plants-13-00576-f005:**
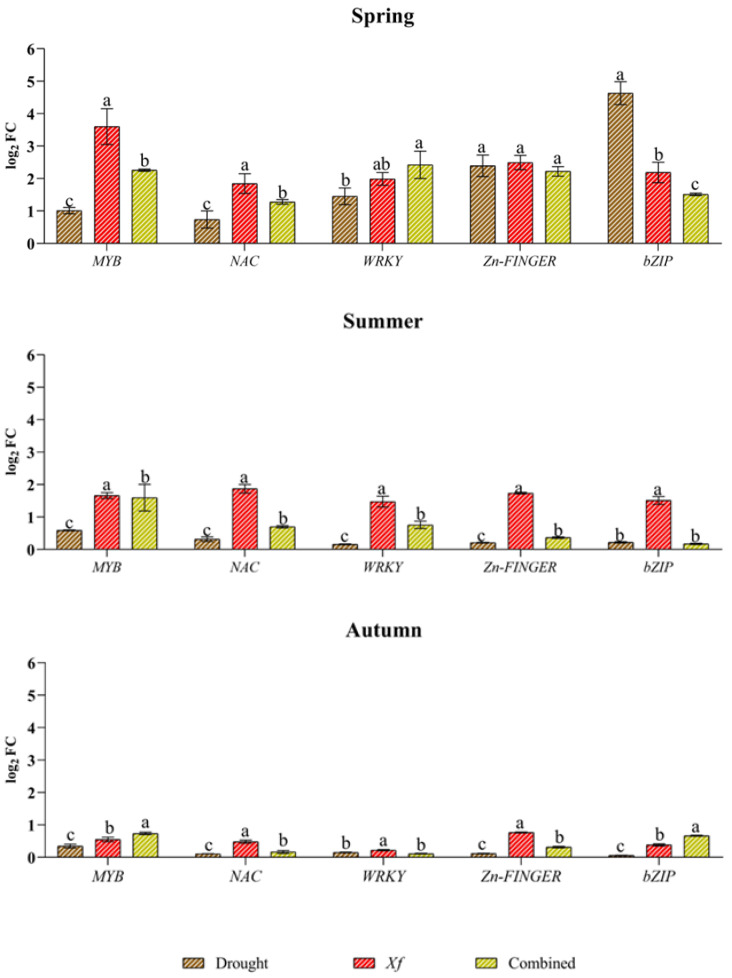
Expression analysis of transcription factors (TFs) involved in the regulation of defence and responses to different stress factors in plants: dehydration-responsive element binding myeloblastosis (MYB); NAC; WRKY; Zn finger; basic leucine zipper (bZIP); in leaves of cv “Filippo Ceo” subjected to stresses: drought, pathogen *Xylella fastidiosa* combination of both (spring, summer and autumn) in two-year observations, expressed as log_2_ fold change (log_2_ FC). One-way ANOVA with Tukey’s HSD post hoc test was used for the statistical analysis. Different letters correspond to statistically different means.

**Table 1 plants-13-00576-t001:** Sequences of primers used in the RT–qPCR analysis.

Target Gene	Forward (5′ to 3′ Sequence)	Reverse (5′ to 3′ Sequence)	Primer Reference
**Dehydrin**	GTACTCTCATGACACCCACAAAACTAC	CCCGGCCCCACCGTAAGCTCCAGTT	[69]
**LEA protein**	GCAAAAGGTAGGGCAAACAG	TGGCTTTGCTTCTTTGGTCT	[69]
**Zn Finger**	ACACAGGCTTCCTCTACTCCATCTTT	GAACCCTCATTCCGAGACATTTATCAG	[69]
**WRKY**	GCCGAGAAATCACCGACTTC	GTTGTCTGAGGCTTGGGTTG	[70]
**PR**	GGAGATGCCTTTGATGTGGGA	AGCTTGAACTCGCCTTCTGG	[71]
**NAC**	GATAACCCAACTACCACTACCAC	GACAACTCCCAGATACCACG	[72]
**b-ZIP**	GGGTTGAAACACCCAAAAGA	GCGATTCGACAACATCCTCT	[73]
**Actin**	CAGATCATGTTTGAGACCTTCAATGT	CATCACCAGAGTCCAGCACAAT	[73]

## Data Availability

The data presented in this study are available on request from the corresponding author.

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
