# Peer review of "A Physiological and Molecular Focus on the Resistance of “Filippo Ceo” Almond Tree to Xylella fastidiosa"

_plants, 2024, doi:10.3390/plants13050576_

Round 1

Reviewer 1 Report

Comments and Suggestions for Authors

De Pascali et al., present a pretty interesting manuscript in which unveil relevant molecular features about resistance to Xf subsp.pauca  as well as drought in almond tree cultivar "Filipo Ceo".  In my criteria, the study is relevant and timely for almond producers In my criteria, the study is relevant and timely for producers, since it becomes a sustainable alternative for the control of the spread of an important phytopathogen bacteria.  Molecular features enrolled in resistance response are unveiled. However there are a minor aspect that must be  addressed in order to  make the article more enriching. 

  Introduction section: 

1. line 67 the expression " for example" Please replace or modify the writing of the paragraph.

2.lines 102-104  .please introduce some importan information about why the almond tree is able to grow in conditions of water shortage without requiring irrigation 

Material and methods 

 subheading:  Total RNA Isolation, cDNA Synthesis, and Real-Time PCR Analysis. Despite that the used techniques are effectively referenced, In order to facilitate the use and reproducibility of these techniques, it is pertinent to introduce a general description on the conditions of   Total RNA Isolation, cDNA Synthesis, and Real-Time PCR Analysis

Comments on the Quality of English Language

The quality of English is good of easy understanding, however there are minor grammar errors that could be remedied

Author Response

Please look at the attached file.

Reviewer 2 Report

Comments and Suggestions for Authors

    All scientific names should have the authority included when first cited in the text

    line 62 The sentence needs to be rephrased, Something is missing.

    line 72 The authors can be deleted.

    lines 122, 143, 180, 207, 252, 253 Scientific names should be in italics

    line 302 Change to Lee et al.

Author Response

Please look at the attached file.

Reviewer 3 Report

Comments and Suggestions for Authors

My comments can be found in the attached document.

Comments on the Quality of English Language

Please correct the grammar in most areas of the manuscript, it is hard to follow due to grammatical errors. In addition, I recommend focusing on a more formal and objective presentation of the findings. My comments can be found in the MS.

Author Response

Please look at the attached file.
